# Impact of Adipose-Derived Mesenchymal Stem Cells (ASCs) of Rheumatic Disease Patients on T Helper Cell Differentiation

**DOI:** 10.3390/ijms23105317

**Published:** 2022-05-10

**Authors:** Ewa Kuca-Warnawin, Magdalena Plebańczyk, Marzena Ciechomska, Marzena Olesińska, Piotr Szczęsny, Ewa Kontny

**Affiliations:** 1Department of Pathophysiology and Immunology, National Institute of Geriatrics, Rheumatology and Rehabilitation, Spartańska 1, 02-637 Warsaw, Poland; magdalena.plebanczyk@spartanska.pl (M.P.); marzena.ciechomska@gmail.com (M.C.); ewa.kontny@spartanska.pl (E.K.); 2Clinic of Connective Tissue Diseases, National Institute of Geriatrics, Rheumatology and Rehabilitation, 02-637 Warsaw, Poland; marzena.olesinska@spartanska.pl (M.O.); piotr.szczesny@spartanska.pl (P.S.)

**Keywords:** systemic lupus erythematosus, systemic sclerosis, adipose-derived mesenchymal stem cells, Th-cell differentiation

## Abstract

Complex pathogenesis of systemic lupus erythematosus (SLE) and systemic sclerosis (SSc) is associated with an imbalance of various Th-cell subpopulations. Mesenchymal stem cells (MSCs) have the ability to restore this balance. However, bone marrow-derived MSCs of SLE and SSc patients exhibit many abnormalities, whereas the properties of adipose derived mesenchymal stem cells (ASCS) are much less known. Therefore, we examined the effect of ASCs obtained from SLE (SLE/ASCs) and SSc (SSc/ASCs) patients on Th subset differentiation, using cells from healthy donors (HD/ASCs) as controls. ASCs were co-cultured with activated CD4^+^ T cells or peripheral blood mononuclear cells. Expression of transcription factors defining Th1, Th2, Th17, and regulatory T cell (Tregs) subsets, i.e., T-bet, GATA3, RORc, and FoxP3, were analysed by quantitative RT-PCR, the concentrations of subset-specific cytokines were measured by ELISA, and Tregs formation by flow cytometry. Compared with HD/ASCs, SLE/ASCs and especially SSc/ASCs triggered Th differentiation which was disturbed at the transcription levels of genes encoding Th1- and Tregs-related transcription factors. However, we failed to find functional consequences of this abnormality, because all tested ASCs similarly switched differentiation from Th1 to Th2 direction with accompanying IFNγ/IL-4 ratio decrease, up-regulated Th17 formation and IL-17 secretion, and up-regulated classical Tregs generation.

## 1. Introduction

Systemic lupus erythematosus (SLE) and systemic sclerosis (SSc) are multisystem, incurable rheumatic diseases of an autoimmune background with life-threatening complications and high mortality rate [1,2]. The pathogenesis of both disorders is complex and not fully understood. Chronic inflammation, the presence of multiple autoantibodies, an unbalanced immune response, and activation of various immune cells, including T lymphocytes, are characteristic features of these diseases [3,4]. Accumulating data show that dysregulations in the differentiation process of CD4^+^ T cells and aberrant T-cell homeostasis are crucial events in SLE [5,6,7,8] and SSc [9,10,11] pathology. Several functionally different subsets of CD4^+^ T helper (Th) cells has been observed. Between them, Th1, Th2, and Th17 cells act as effector cells and are thought to contribute to the development of various autoimmune diseases, while regulatory T cells (Tregs) play a protective role [12,13]. In SLE, an imbalance between Th1/Th2 and Th17/Tregs subsets with skewing towards Th1 [6,14,15] and Th17 responses [16,17,18,19], along with accompanying impairment of Tregs [16,17,18,20,21,22,23,24], have been reported. In SSc, a higher frequency of Th2 and Th17 cells, while the decreased number of Th1 [9,10,25,26,27,28,29,30] and Tregs [11,31] have been shown, indicating skewing of T cell differentiation towards Th2 and Th17 responses.

It is proposed that recovery of the immune balance between Th effector subsets and Tregs offers a promising therapeutic option for these patients [7,32,33]. Accumulating and encouraging evidence suggests that clinical application of mesenchymal stem cells (MSCs) may represent such a therapeutic method since these cells exert the immunoregulatory effects on CD4^+^ T-cell homeostasis by promoting the generation of functional Tregs, inhibiting differentiation into Th1 and Th17 subsets, and modulating Th2 response [34,35]. Mesenchymal stem cells can be isolated from various adult tissue, including bone marrow (BM) and adipose tissue. Unfortunately, SLE bone marrow-derived MSCs (BM-MSCs) exhibit many abnormalities and functional defects [36,37,38]. Furthermore, data concerning the biology of BM-MSCs of SSc patients are inconsistent, showing these cells’ defects or similarities to BM-MSCs of healthy donors in terms of phenotype properties and biological functions, including immunomodulatory potential [39,40]. Therefore, primarily allogeneic MSCs, originating from BM of healthy donors or umbilical cord (UC-MSCs) or other perinatal tissues, are used in clinical trials [40,41]. Although most of these trials indicate clinical improvement in SLE and SSc patients [37,39,41], some emphasize little scientific evidence about allogeneic MSCs transplantation benefits [37,42]. Because MSCs express low levels of major histocompatibility complex (MHC) class II molecules, they are believed to be immune-privileged. However, some observations suggest that allogeneic MSCs can induce an adaptive immune response under appropriate conditions, followed by an adverse response [40]. Furthermore, the long-term consequences of transplantation of allogeneic MSCs are not yet known [41].

Adipose-tissue-derived MSCs (ASCs) share similar morphological, phenotypical, and functional characteristics to BM-MSCs and UC-MSCs, and thus may represent an appropriate alternative source of MSCs for treatment. Importantly, ASCs have several advantages over MSCs derived from other tissues, e.g., less invasive and repeatable harvesting, lower senescence, higher cell proliferation rates, genetic stability, and the most potent immunosuppressive capacity against immune cells including T and B lymphocytes and monocytes [40]. Furthermore, because in autoimmune diseases, chronic inflammation seems to be responsible for BM-MSCs impairment [40], and the inflammatory response in BM and the inflamed fat pad is of a higher grade than in peripheral adipose tissue [43,44], one may speculate that biological properties of ASCs from subcutaneous fat are better preserved than those derived from bone marrow. However, the biological features of ASCs of SLE (SLE/ASCs) and SSc (SSc/ASCs) patients are poorly characterised, and especially, data on the immunoregulatory functions of these cells are scarce. In addition, it has been demonstrated that SSc/ASCs exhibit decreased proliferation rate, metabolic activity, migration potential [39], and may contribute to pathological fibrosis through the adipocyte-to-myofibroblast transition process [45].

Nevertheless, several preclinical and clinical research studies have shown the beneficial effects of autologous ASCs-based local treatment for facial and hand cutaneous manifestations in SSc patients [45]. Moreover, in murine lupus models, systemic application of human ASCs was reported to bring beneficial therapeutic and immunomodulatory effects [37]. In addition, we have previously reported decreased basal levels of specific surface molecules on both SLE/ASCs and SSc/ASCs and some changes in the secretory activity of these cells that altogether suggested their functional disorders [46]. Despite this, we have recently found that SLE/ASCs and SSc/ASCs retain the standard capability to regulate activation and expansion of allogeneic T cells, act by similar mechanisms as ASCs of healthy donors, and thus may have therapeutic value [47,48].

From the perspective of the potential systemic application of autologous ASCs to treating SLE and SSc patients, it is essential to clarify whether these cells have similar biological properties as ASCs of healthy donors. Therefore, to further characterise the immunoregulatory properties of SLE/ASCs and SSc/ASCs, in the present study, we investigated the impact of these cells on the differentiation of allogeneic Th cells, using ASCs lines from healthy donors (HD/ASCs) as the reference. For this purpose, the expression of lineage-defining transcription factors of Th1, Th2, Th17, and Treg subsets (T-bet, GATA3, RORc, and FoxP3, respectively), the secretion of subset specific cytokines (IFNγ, IL-4, IL-17), and formation of classical Tregs (CD4^+^CD25^high^FoxP3^+^) were examined in the co-cultures of ASCs with activated target cells. In addition, as the target cells, purified CD4^+^ cells and peripheral blood mononuclear cells (PBMCs) were used to evaluate the direct impact of ASCs on Th cell differentiation and the possible contribution of other cells in this process, respectively. Our results show that ASCs from SLE and SSc patients, similar to HD/ASCs, modify Th differentiation and lineage-associated cytokine secretion, shifting it from Th1 to Th2 and up-regulating generation of both Th17 and classical Treg cells. However, the differentiation process triggered by ASCs of patients shows some abnormalities at the transcriptional levels, but functional consequences of this anomaly, if any, require further study.

## 2. Results

### 2.1. Patients

The demographic and clinical data of SLE and SSc patients are shown in Table 1.

Most patients presented low disease activity, and only four SLE and two SSc patients had active disease. Among SSc patients, 47% presented limited, while 53% had a systemic form of the disease. All patients had antinuclear antibodies (ANA), and SSc patients had higher ANA titer. The majority of patients also had disease-specific autoantibodies. SLE patients were characterised by higher proteinuria. All patients were treated with immunosuppressive drugs, and none received biological therapy.

### 2.2. Modulatory Effects of ASCs on the Expression of Th Lineage-Defining Transcription Factors

Activation of separately cultured purified CD4^+^ T lymphocytes via anti-CD3/CD28 pathway (Tact cells) up-regulated the expression of T-bet mRNA and down-regulated GATA3 mRNA, resulting in a significant increase of T-bet/GATA3 ratio and thus a shift toward Th1 response, compared with resting CD4^+^ T cells (Appendix A). The addition of TI-treated (TNF + IFNγ) and untreated HD/ASCs to Tact cells triggered opposite effects, i.e., significantly enhanced GATA-3 mRNA levels, leaving T-bet mRNA levels unchanged, thus finally reducing T-bet/GATA3 ratio and causing a shift toward Th2 response (Figure 1A). Similar effects were observed in the co-cultures of Tact cells with both SLE/ASCs (Figure 1B) and SSc/ASCs (Figure 1C). However, an impact of these cells, especially SSc/ASCs, was less pronounced and reduction of T-bet/GATA3 was weaker, compared with HD/ASCs exerted effects. Moreover, in control separately cultured Tact cells, the RORc/FoxP3 ratio was significantly higher than in resting CD4^+^ T cells, resulting in a shift toward Th17 response, caused by a tendency to increase RORc and decrease of FoxP3 mRNAs levels (Appendix A). In the presence of HD/ASCs, further up-regulation of both RORc and FoxP3 mRNAs levels and RORc/FoxP3 ratio was observed (Figure 1D), indicating a shift toward Th17 response. In comparison with HD/ASCs, the presence of SSc/ASCs and SLE/ASCs triggered a similar increase of FoxP3 mRNA but weaker up-regulation of RORc mRNAs (Figure 1E,F). Furthermore, SSc/ASCs failed to increase RORc mRNAs significantly. In the co-cultures of Tact with patients’ ASCs, RORc/FoxP3 ratio had decreasing tendency and in the presence of SLE/ASCs was significantly lower, compared with HD/ASCs containing co-cultures. Thus, in this experimental setting ASCs from both tested groups of patients had a weaker ability to up-regulate RORc mRNA. SSc/ASCs were also a weaker inducer of GATA3 in activated CD4^+^ T cells.

In comparison with resting PBMCs, in PBMCs stimulated with PHA (PBMCs_PHA_), there was a decrease of mRNAs levels encoding tested transcription factors, except RORc, which raised significantly, and neither T-bet/GATA3 nor RORc/FoxP3 ratios change significantly (Appendix A). The expression T-bet mRNAs and T-bet/GATA3 ratio were significantly higher in Tact than in PBMC_PHA_ (Appendix A). When HD/ASCs were co-cultured with PBMCsPHA, the changes in the expression of transcription factors mRNAs were similar as in co-cultures containing Tact cells as target cells (compare Figure 1A,B and Figure 2A,B), i.e., up-regulation of GATA3, RORc, and FoxP3, but stable T-bet level. By contrast, in the co-cultures of PBMCs_PHA_ with ASCs from tested patients, no significant changes in the expression of transcription factors were found. In some cases, i.e., GATA3 and/or FoxP3 mRNAs for co-cultures containing SSc/ASCs (Figure 2C,F) and SLE/ASCs (Figure 2B,E), the expression levels were lower than in HD/ASCs + PBMCs_PHA_ co-cultures. In this experimental setting, the ratio of T-bet/GATA3 and RORc/FoxP3 remained unchanged. These results showed the weaker ability of ACSs, especially those from tested patients, to modulate the expression of Th subset-related transcription factors in mitogen-activated PBMCs.

### 2.3. The Effects of ASCs on the Release of Th Subset-Specific Cytokines

Both types of activated target cells secreted more IFNγ, IL-4, and IL-17AF than respective resting cells, and Tact produced more IFNγ and IL-4 than PBMCs_PHA_, while IL-17AF secretion did not differ significantly between them (Appendix A).

In the presence of HD/ASCs and SLE/ASCs, the secretion of IFNγ by Tact cells remained unchanged, while IL-4 release was increased (Figure 3A,B). Conversely, in the co-cultures containing TI-treated or both TI-treated and untreated SSc/ASCs, reduction of IFNγ and stable levels of IL-4, respectively, were observed (Figure 3A,B). Analysis of cytokine response in individual experiments showed that both HD/ASCs and ASCs from tested patients exert a modulatory effect on IFNγ and IL-4 secretion (Figure 4A,B), up-regulating these cytokines when the basal secretion is low and down-regulating them when the basal level is high. This modulatory impact of ASCs resulted in a significant reduction of the IFNγ/IL-4 ratio (Figure 4C). When PHA-activated PBMCs were used as the target cells, both HD/ASCs and ASCs from tested patients exerted a significant inhibitory effect on IFNγ secretion (Figure 3C and Figure 4D). In the co-cultures of ASCs with PBMCs_PHA_, the release of IL-4 was very low, usually below the detection limit (<5 pg/mL); therefore, we excluded these results from analysis (data not shown).

As shown in Figure 5, in the co-cultures of ASCs with both types of target cells, the secretion of IL-17AF was significantly up-regulated in the presence of ASCs from healthy donors as well as from the patients. This increase was seen in almost every experiment, and there was no significant difference between co-cultures with HD/ASCs versus (vs.) SLE/ASCs and SSc/ASCs.

### 2.4. Modulatory Effects of ASCs on the Generation of Tregs

Compared with resting cells, activation of both target cells significantly up-regulated the number of Tregs, and this increase was higher in the case of Tact cells than in PBMCs_PHA_ (Appendix A). Compared with separately cultured _act_CD4^+^ T cells, the generation of classical Tregs (CD4^+^CD25^high^FoxP3^+^) was reduced when CD4^+^ T cells were co-cultured with both untreated and TI-pre-stimulated HD/ASCs and SSc/ASCs. At the same time, only decreasing tendency was observed in the presence of SLE/ASCs (Figure 5A). However, opposite effects were observed in the co-cultures of ASCs with PHA-activated PBMCs (Figure 5B). In these conditions, there was a significant increase in the number of Tregs compared to activated PBMCs alone (Figure 5B). There was no difference in the generation of Treg cells between co-cultures containing HD/ASCs and SLE/ASCs, while TI-pre-treated SSc/ASCs were less able to up-regulate the number of these cells. The FoxP3 mean fluorescence intensity (MFI) values remained unchanged in the co-cultures of ASCs with both types of target cells, except for some decrease noted in the co-cultures of CD4^+^ T cells with SSc/ASCs (Figure 5C,D).

## 3. Discussion

T lymphocytes are accountable for balancing diverse regulatory and effector immune functions. To accomplish this task, activated CD4^+^ Th cells differentiate into adequate subsets that are functionally tailored to eliminate existing danger signals, such as Th1, Th2, and Th17, or Tregs that prevent excessive immune/inflammatory response. T-cell differentiation into a particular Th subset is controlled by a definite panel of cytokines present in the surrounding environment and master transcription factors. Among others, T-bet, GATA3, RORc, and FoxP3 define the generation of Th1, Th2, Th17, and regulatory T (Tregs) cells. These transcription factors control, in turn, the expression of Th1, Th2, and Th17 specific cytokines, i.e., IFNγ, IL-4/IL-15/IL-13, and IL-17, whereas FoxP3 coordinates Tregs development but represses the generation of other Th subsets [12,49,50,51]. In SLE and SSc, the homeostasis of Th1, Th2, Th17, and Treg cells and a balance between the subset-specific cytokine production that are essential for maintaining human health are highly disturbed. There is a skewing in PBMCs of SLE patients towards either Th1 [6,14,15] or less often to Th2 [52] response. Additionally, in the kidneys of patients with lupus nephritis (LN), one of the most worrying SLE manifestations, Th1, predominate over Th2 cells [14,53,54]. In addition, both in PBMCs and in disease-affected organs of SLE patients, increased numbers and over-activation of Th17 cells [16,17,18,19], accompanied by quantitative and qualitative impairments of Tregs [16,17,18,20,21,22,23,24], have been frequently documented. In the circulation and affected organs of SSc patients, higher frequency of Th2 and Th17 cells, but the decreased number of Th1 [9,10,25,26,27,28,29,30] and Tregs, that also show functional defects [11,31], have been documented, indicating skewing of T cell differentiation towards Th2 and Th17 response.

It is well documented that MSCs can affect the balance of the CD4. Th subsets both in vitro and in vivo. In animal models of various human diseases, application of ASCs has a therapeutic benefit. Intriguingly, in Th-2 mediated diseases, clinical improvement is accompanied by immunomodulation from a Th2 to a Th1-biased response [55], while the opposite effect is observed in Th1-mediated disorders [56,57]. In murine models of lupus and bleomycin-induced mice scleroderma, transplantation of ASCs improved clinical symptoms and concomitantly reduced the number of Th1 and/or Th17 cells but increased the number of Tregs [58,59,60]. These observations emphasise that, like MSCs derived from various tissues [61], ASCs are endowed with immunomodulatory activity, which is highly plastic in response to disease context. However, it is unknown whether ASCs from SLE and SSc patients possess such capabilities. The results of in vitro research confirm the functional plasticity of ASCs, displaying those various types of stimuli and cellular context create a specific milieu that exerts different effects on MSCs and, in turn, changes their impact on immune cells [62]. In accordance with these in vivo and in vitro data, the present results show similarities and differences in the influence of ASCs on the expression of Th subset-specific transcription factors (Figure 1 vs. Figure 2) and cytokines (Figure 3 and Figure 4) as well as Tregs generation (Figure 6) in the co-cultures with different types of activated target cells. We observed that activation of CD4^+^ T cells via the anti-CD3/CD28 pathway shifted the domination of transcription factors expression towards Th1 (Appendix A), whereas the presence of HD/ASCs switched it from Th1 to Th2 (Figure 1A). Consistently, the secretion of IFNγ and IL-4 were modulated, leading to the significant decrease of the IFNγ/IL-4 ratio (Figure 4A–C). The changes in PBMCs response were slightly less visible because, in the presence of HD/ASCs, there was only a tendency to shift towards Th2 differentiation (Figure 2A). However, a significant reduction of Th1- specific IFNγ release was also found (Figure 4D), supporting similar drift as in the case of activated CD4^+^ T cells. Similar to other observations [63], we noticed a discrepancy between T-bet mRNA levels and IFNγ production. It is likely that soluble factors, able to influence IFNγ production without altering T-bet expression, such as PGE_2_, can be responsible for it [64].

Moreover, in the presence of HD/ASCs, the expression of Th17-related RORc (Figure 1D vs. Figure 2D) and IL-17AF secretion (Figure 5A,C vs. Figure 5B,D) were significantly enhanced in both types of activated target cells, showing that this effect is exerted directly by ASCs and not modified by other cell types present in PBMCs pool. Data concerning the impact of ASCs on the generation of Th17 cells and the production of IL-17 are controversial. Depending on in vitro culture conditions, both inhibitory [34,65,66] and promoting effects [65,67,68,69] were observed. The present results are consistent with the reports showing up-regulation of Th17 and suppression of Th1 response in CD4^+^ T cells or PBMCs, caused by the direct action of ASCs or indirect action of human BM-MSCs via a myeloid cell-mediated mechanism, respectively [70,71]. Regarding HD/ASCs, the capability of SLE/ASCs to shift the transcription factors expression in activated target cells towards Th2 and to up-regulate RORc levels were weaker (in CD4^+^ T cells) if any (in PBMCs), whereas SSc/ASCs rather failed to modify these transcription factors expression in target cells (Figure 1 and Figure 2). Despite this, patients’ ASCs reduced IFNγ/IL-4 ratio and IFNγ secretion as well as up-regulated IL-17AF release similarly as HD/ASCs (Figure 4 and Figure 5). Thus, the capability of patients’ ASCs to modify the effector function of Th1, Th2, and Th17 seems to be expected and, in agreement with our previous report [68], results from the direct action of ASCs on T cells because similar effects were observed in ASCs co-cultures with both target cell types. It is worth underlying that ASCs-triggered switch from Th1 to Th2 is favourable in the context of SLE pathology but rather disadvantageous when it comes to SSc. Fortunately, observed weak, if any, enhancing effect of SSc/ASCs on both GATA3 mRNA expression and IL-4 production argue in favour of beneficial effects of these cells. By contrast, the promotion of Th17 cells and IL-17AF production is an unfavourable feature of ASCs.

As for ASCs impact on Tregs generation, our results showed differences between the response of CD4^+^ T and PBMCs (Figure 1, Figure 2 and Figure 6). Although HD/ASCs and ASCs from tested patients significantly up-regulated the expression of Treg lineage-specifying transcription factor, FoxP3, in co-cultured CD4^+^ T cells, this was not followed by an increase in the number of classical (CD4^+^CD25^high^FoxP3^+^) Tregs. Conversely, in these conditions, either decreased or unchanged number of Tregs was stated, showing the divergence between the above events of Tregs generation. On the other hand, only HD/ASCs, but neither SLE/ASCs nor SSc/ASCs, up-regulated FoxP3 mRNA levels in PBMCs target cells. However, all tested ASCs types significantly enhanced the number of Tregs in co-cultures, indicating full compatibility between FoxP3 expression and Tregs number only in the case of HD/ASCs. It is known that apart from FoxP3, which orchestrate the development and function of Tregs, other transcription factors also contribute to Treg cell lineage specification. For example, Foxp1 regulates the expression of critical molecules implicated in Tregs suppressor function, such as CD25 and CTLA-4, while Foxo transcription factors directly control *FoxP3* and *CTLA4* genes [51]. Thus, it is possible that T-cell differentiation into Tregs lineage triggered by patients’ ASCs is somehow disturbed at the transcriptional level. However, it does not disturb the generation of phenotypically specified Tregs. Whether these cells are fully functional suppressor cells requires further studies. Nevertheless, our results are in line with the report showing the ability of human BM-MSCs to increase Tregs generation only indirectly by skewing differentiation of monocytes towards anti-inflammatory type 2 macrophages that are essential for Tregs generation [72]. The other data confirm the capability of also ASCs to up-regulate Treg generation in co-cultured PBMCs [68,73,74,75] and to expand Tregs when CD4^+^ T cells are cultured in Treg polarising conditions [76]. Importantly, even though ASCs from patients cannot enhance FoxP3 expression levels in PBMCs target cells, their capability to promote Tregs generation is compatible with ASCs of healthy donors, advocating for their beneficial action in these diseases.

It should be mentioned that both HD/ASCs and ASCs of patients exerted similar modulatory effects on Th differentiation and tested cytokine production regardless of whether cytokine (TI) were primed or not. It is consistent with reports showing that ASCs do not require priming with pro-inflammatory cytokines to efficiently up-regulate RORc expression in target cells [77]. However, it opposes other immunoregulatory functions of these cells, e.g., secretory activity and anti-proliferative action [46,48,78]. In addition, our results have a practical tone, indicating that to assess the immunomodulatory function of ASCs from SLE and SSc patients, it is better to study the effector phases of Th differentiation process than analyse it at the transcriptional level and to use PBMCs rather than CD4^+^ T cells as the target cells.

Our study has some limitations. The first is the lack of functional evaluation of Tregs generated upon co-culture with ASCs. Secondly, we cannot exclude the impact of patients’ treatment on tested functions of their ASCs. However, it is likely that drug-related effects have vanished along with a month-long expansion of these cells in vitro, required to obtain cell numbers enough to perform presented experiments.

## 4. Materials and Methods

### 4.1. Patients and Sample Collection

Two groups of patients, who fulfilled the criteria for SLE (*n* = 19) [79] or SSc (*n* = 19) [80], were included in the study (Table 1). This study met all criteria in the Declaration of Helsinki. The Ethics Committee approved the National Institute of Geriatrics, Rheumatology, and Rehabilitation, Warsaw, Poland (the approval protocol no: KBT-8/4/20016). Before enrolment, all patients gave their written informed consent.

### 4.2. ASCs Isolation and Culture

Subcutaneous abdominal fat tissue procurement from SLE or SSc patients, tissue processing, and isolation of ASCs was performed according to the previously described protocol [81]. Five earlier characterised [46], human ASCs lines (Lonza Group, Lonza Walkershille Inc., Walkersville, MD, USA) were used as a reference. All experiments were performed using ASCs at 3–5 passages. ASCs were cultured in a complete culture medium composed of DMEM/F12 (PAN Biotech UK Ltd., Wimborne, UK), 10% foetal calf serum (FCS) (Biochrom, Berlin, Germany), 200 U/mL penicillin, 200 µg/mL streptomycin (Polfa Tarchomin S.A., Warsaw, Poland), and 5 µg/mL plasmocin (InvivoGen, San Diego, CA, USA). In addition, for some experiments, ASCs were stimulated for 24 h with human recombinant tumor necrosis factor (TNF) and interferon γ (IFNγ) (both from R&D Systems, Minneapolis, MN, USA; each applied at 10 ng/mL).

### 4.3. Cell Co-Cultures

All cell co-cultures were performed in the complete culture medium (see above). ASCs (1 × 10^4^/well/0.2 mL of medium) were seeded into 96-well plates, stimulated with IFNγ and TNF (ASCs_TI_), or were left untreated. PBMCs were isolated from buffy coats obtained from healthy male donors (<60 years old), using Ficoll-Paque (GE Healthcare, Uppsala, Sweden) and routinely applied procedure. The CD3^+^CD4^+^ cells were isolated from PBMCs using EasySep™ Human CD4^+^ T Cell Isolation Kit (Stemcell Technologies, Vancouver, BC, Canada). After isolation, PBMCs or CD4^+^ T cells (1.2 × 10^6^/well/2 mL of medium) were seeded either directly (contacting co-cultures) or on a 0.4 μm pore size Transwell filters (MD24 with carrier for inserts 0.4 MY, Thermo Fisher Scientific, Waltham, MA, USA) (non-contacting co-culture) into 24-well plates with adherent ASCs (5 × 10^4^/well/2 mL of medium). Then, PBMCs were treated with 2.5 µg/mL of phytohaemagglutinin (PHA, Sigma-Aldrich, St. Louis, MO, USA), whereas CD4^+^ T cells were activated with Dynabeads™ Human T-Activator CD3/CD28 (Thermo Fisher Scientific, Waltham, MA, USA). After 5 days of co-culture, PBMCs or CD4^+^ T cells were harvested for cytometric analysis and/or RNA isolation. The concentrations of soluble factors were measured in collected culture supernatants. Separately cultured ASCs, PBMCs, and CD4^+^ T cells were used as the co-culture controls.

### 4.4. Flow Cytometry Analysis

For the CD4^+^ cells or PBMCs analysis, harvested cells were re-suspended in 50 μL of fluorescent-activated cell sorting buffer and stained for 30 min on ice for respective membrane antigens, using fluorochrome-conjugated monoclonal antibodies specific for human CD4–FITC and CD25-PE (both from (BD Pharmingen, San Diego, CA, USA). Cells were fixed and permeabilised in the next step using the FoxP3/transcription factor Staining Buffer Set (Thermo Fisher Scientific, Waltham, MA, USA). Subsequently, intracellular staining using anti FoxP3-APC antibody were performed. After the washing, cells were acquired and analysed using a FACS Canto cell sorter/cytometer and Diva software. Appropriate isotype controls were used in all experiments.

### 4.5. Enzyme-Linked Immunosorbent Assays (ELISAs)

The concentrations of cytokines were measured in culture supernatants in duplicates by specific ELISAs. IL-17AF and IFNγ concentrations were measured using Ready-SET-Go kits (eBioscience, San Diego, CA, USA), while IL-4 using Human IL-4 Uncoated ELISA kit (Thermo Fisher Scientific, Waltham, MA, USA).

### 4.6. Polymerase Chain Reaction (PCR)

RNeasy Mini Kit was used for mRNA isolation according to the manufacturer’s protocol (Qiagen, Hilden, Germany). High-Capacity cDNA Reverse Transcription Kit was used to reverse transcribe RNA to cDNA (Thermo Fisher Scientific, Waltham, MA, USA). The 10 µL PCR reaction included 2 µL RT product, 5 µL TaqMan Gene expression master Mix, 0.5 µL probe mix of the TaqMan (T-bet, GATA, RORc, FoxP3, and following housekeeping genes GAPDH, TBP, RPL13a—all from Thermo Fisher Scientific, Waltham, MA, USA), and 2.5 µL of water (Genoplast, Rokocin, Poland). Reactions were performed at 50 °C for 2 min, 95 °C for 10 min, followed by 50 cycles at 95 °C for 15 s and 60 °C for 1 min. Samples were analysed in triplicate using the QuantStudio 5 qRT-PCR machines (Thermo Fisher Scientific, Waltham, MA, USA). Gene expression was evaluated using ΔΔCT-method.

### 4.7. Statistical Analysis

Data were analysed using GraphPad PRISM software version 7. The Shapiro-Wilk test was used as a normality test. One-way analysis of variance (ANOVA) with repeated measures and post-hoc Tukey test was used to assess the effect of untreated and TNF/IFNγ (TI)-treated ASCs on target cells, and to compare cell contacting versus (vs.) non-contacting co-cultures. The differences between ASC lines from healthy donors (HD/ASCs) and ASCs from SLE (SLE/ASCs) and SSc (SSc/ASCs) patients were analysed using the Kruskal-Wallis and Dunn’s multiple comparison tests. Probability values less than 0.05 were considered significant.

## 5. Conclusions

We report that ASCs from healthy donors and SLE and SSc patients exert comparable immunomodulatory impact on the final phase of the allogeneic CD4^+^ T-cell differentiation process. All tested ASCs can: (i) switch this process from Th1 to Th2 direction with accompanying IFNγ/IL-4 ratio decrease, (ii) up-regulate Th17 formation and IL-17 secretion, and (iii) up-regulate classical Tregs generation. Despite this, ASCs from SLE and especially SSc patients triggered Th differentiation, which is somehow disturbed at the transcription levels of genes encoding Th1 and Tregs lineage-specific transcription factors. However, it is unknown whether this abnormality has any functional consequences. We also confirm that Th1/Th2 switch results from the direct action of ASCs, whereas up-regulation of Tregs is indirect, mediated by other cell types present in the PBMCs pool.

## Figures and Tables

**Figure 1 ijms-23-05317-f001:**
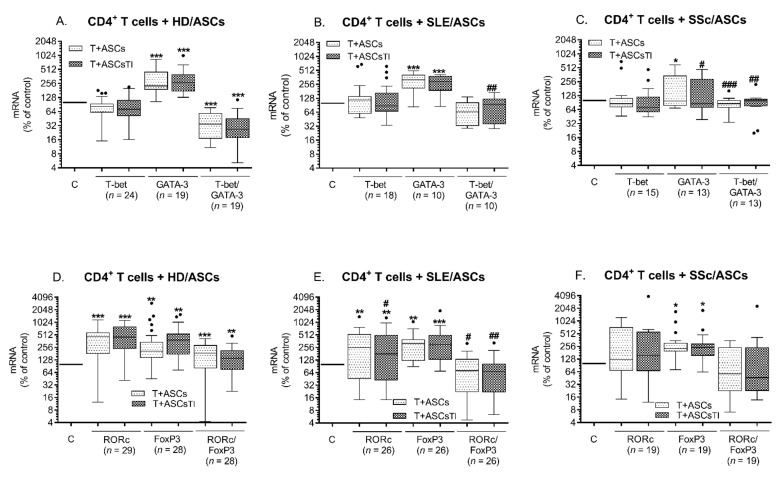
Direct effects of ASCs on the expression of Th subset-specific transcription factors in activated T cells. T helper (CD3^+^CD4^+^) lymphocytes (T) were isolated from peripheral blood of 22 healthy volunteers, activated via CD3/CD28 pathway, and co-cultured for 5 days with either untreated or TNF + IFNγ (TI) pre-stimulated ASCs lines from healthy donors (HD/ASCs, **A**,**D**), SLE (SLE/ASCs, **B**,**E**), or SSc (SSc/ASCs, **C**,**F**) patients. The co-cultures were done using a random combination of both cell types, and the number of experiments performed is shown (n). The levels of expression of mRNAs encoding transcription factors specific for Th1, Th2, Th17, and Treg subsets (T-bet, GATA3, RORc, and FoxP3, respectively) were assessed by quantitative RT-PCR, as described in the Materials and methods. Results are expressed as the percentage of the mRNAs expression levels regarding those found in separately cultured control activated T cells and are shown as the Tukey’s boxes with the median (horizontal line), interquartile range (IQR, box), lower and upper whiskers (data within 3/2xIQR), and outliers (points/dots). One-way analysis of variance (ANOVA) with repeated measures and post-hoc Tukey test was used to evaluate the effect of ASCs (T vs. T + ASCs or ASCs_TI_; * *p* = 0.05–0.01; ** *p* = 0.01–0.001; *** *p* = 0.001–0.0001). The inter-group (HD vs. SLE vs. SSc) comparison was performed using the Kruskal-Wallis and Dunn’s multiple comparison test (# *p* = 0.05–0.01; ## *p* = 0.01–0.001; ### *p* = 0.001–0.0001 for the groups of patients vs. HD comparison).

**Figure 2 ijms-23-05317-f002:**
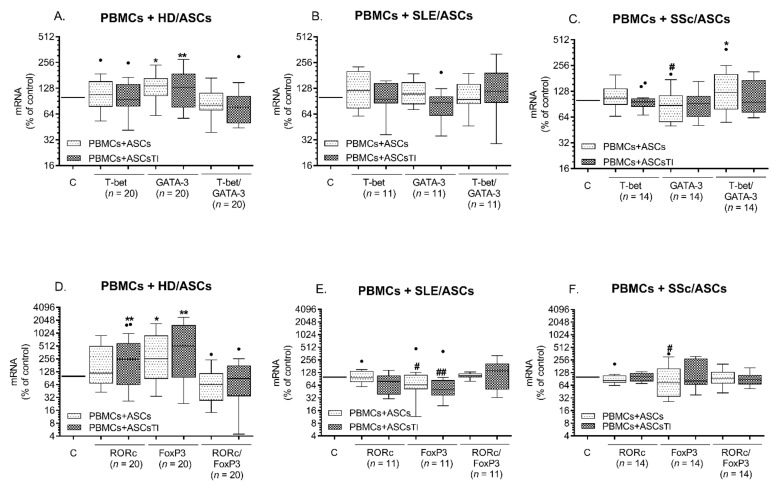
Changes in the expression of Th subset-specific transcription factors in mitogen-activated PBMCs co-cultured with ASCs. Explanations as in Figure 1, except that HD/ASCs (**A**,**D**), SLE/ASCs (**B**,**E**), and SSc/ASCs (**C**,**F**) lines were co-cultured with PHA-stimulated PBMCs that were isolated from peripheral blood of nine healthy volunteers. The ASCs + PBMCs co-cultures were done using a random combination of both cell types, and the number of experiments performed is shown (n). Results are expressed as the percentage of the mRNAs expression levels regarding those found in separately cultured control activated PBMCs (**C**) and are shown as the Tukey’s boxes; n—the number of experiments performed. * *p* = 0.05–0.01; ** *p* = 0.01–0.001 for intra-group comparison (PBMCs vs. PBMCs + ASCs or ASCs_TI_); # *p* = 0.05–0.01; ## *p* = 0.01–0.001 for inter-group comparison (HD vs. SLE vs. SSc).

**Figure 3 ijms-23-05317-f003:**
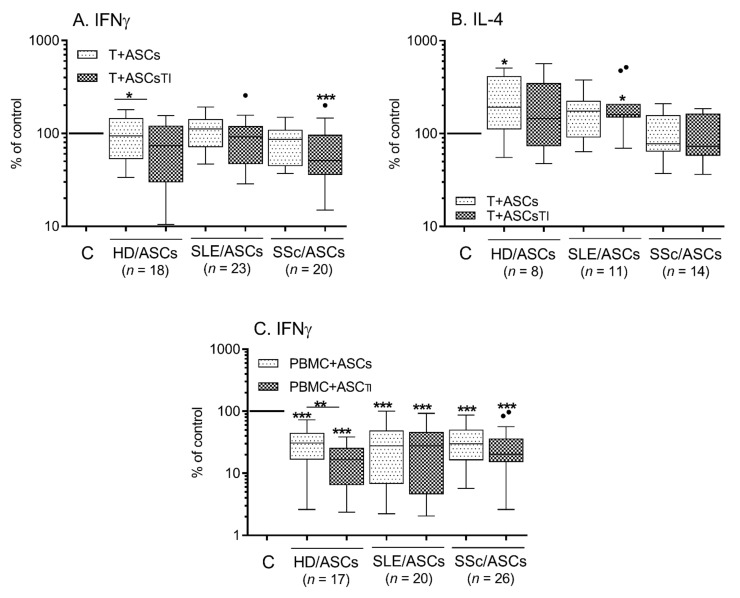
The effects of ASCs on the secretion of Th1 and Th2 subset-related cytokines by activated target cells. Cells were prepared and co-cultured as described in Figure 1 and Figure 2, except that HD/ASCs (*n* = 5), SLE/ASCs (*n* = 16), and SSc/ASCs (*n* = 14) lines were co-cultured with activated T cells (**A**,**B**) or activated PBMCs (**C**) that were isolated from peripheral blood of 15 and 17 healthy volunteers, respectively. The concentrations of IFNγ and IL-4 were measured in culture supernatants by specific ELISAs as described in the Materials and methods. Results are expressed as the percentage of the cytokine levels regarding those found in separately cultured activated control target cells (**C**) and are shown as the Tukey’s boxes; n—the number of experiments performed. * *p* = 0.05–0.01; ** *p* = 0.01–0.001; *** *p* = 0.001–0.0001 for intra-group comparison; the inter-group differences were statistically insignificant.

**Figure 4 ijms-23-05317-f004:**
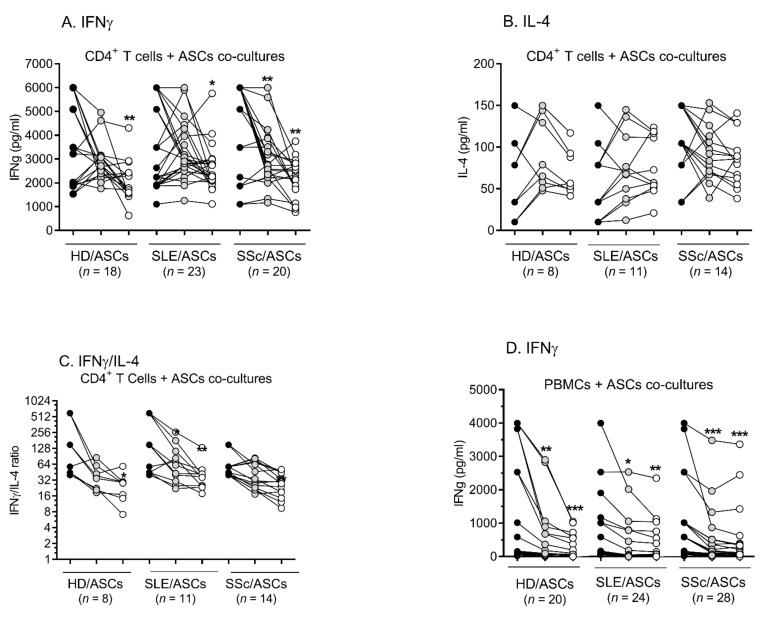
The production of IFNγ and IL-4 in individual experiments. Cell preparation, culture conditions, and the explanations as described in Figure 3. Lines between points identify cultures containing the same combination of ASCs and activated T cells (**A**–**C**) or activated PBMCs (**D**). Data are shown as the cytokine concentrations (**A**,**B**,**D**) or the IFNγ/IL-4 ratio (**C**). Black circles—separately cultured activated control target cells (_act_T cells or PBMCs_PHA_); grey circles—activated target cells + ASCs; unfilled circles—activated target cells + ASCs_TI_. * *p* = 0.05–0.01; ** *p* = 0.01–0.001; *** *p* = 0.001–0.0001 for intra-group comparison; the inter-group differences were statistically insignificant.

**Figure 5 ijms-23-05317-f005:**
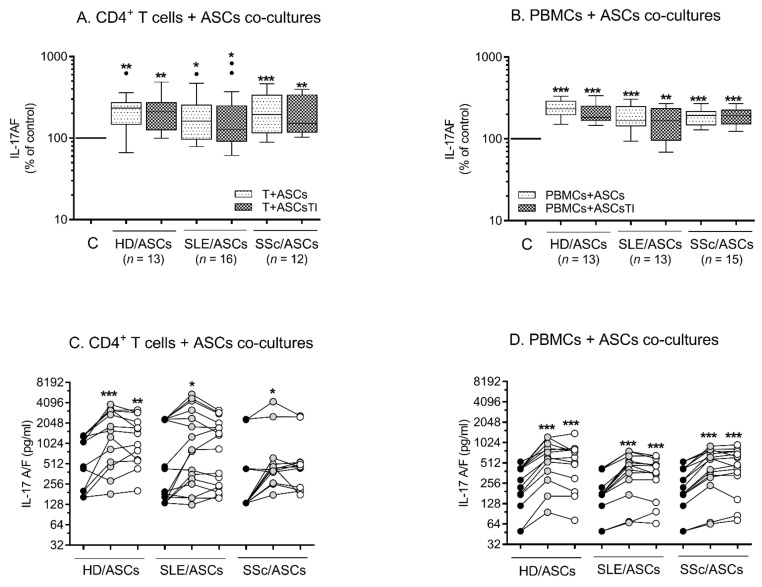
The effects of ASCs on the secretion of IL-17AF by activated target cells. Cell preparation and culture conditions as in Figure 3 and Figure 4, except that HD/ASCs (*n* = 5), SLE/ASCs (*n* = 18), and SSc/ASCs (*n* = 17) lines were co-cultured with activated T cells or activated PBMCs that were isolated from peripheral blood of 13 healthy volunteers. The concentrations of IL-17AF were measured in culture supernatants by specific ELISAs as described in the Materials and methods. (**A**,**B**)—the results are expressed as the percentage of the cytokine levels regarding those found in separately cultured activated control target cells (**C**) and are shown as the Tukey’s boxes. (**C**,**D**)—the same data shown as IL-17 AF concentrations in individual experiments in which lines between points identify cultures containing the same combination of ASCs and T cells (**A**,**C**) or PBMCs (**B**,**D**); *n*—the number of experiments performed. Black circles—separately cultured activated T cells (**C**) or PBMCs (**D**); grey circles—activated target cells + ASCs; unfilled circles—activated target cells + ASCs_TI_. * *p* = 0.05–0.01; ** *p* = 0.01–0.001; *** *p* = 0.001–0.0001 for intra-group comparison; the inter-group differences were statistically insignificant.

**Figure 6 ijms-23-05317-f006:**
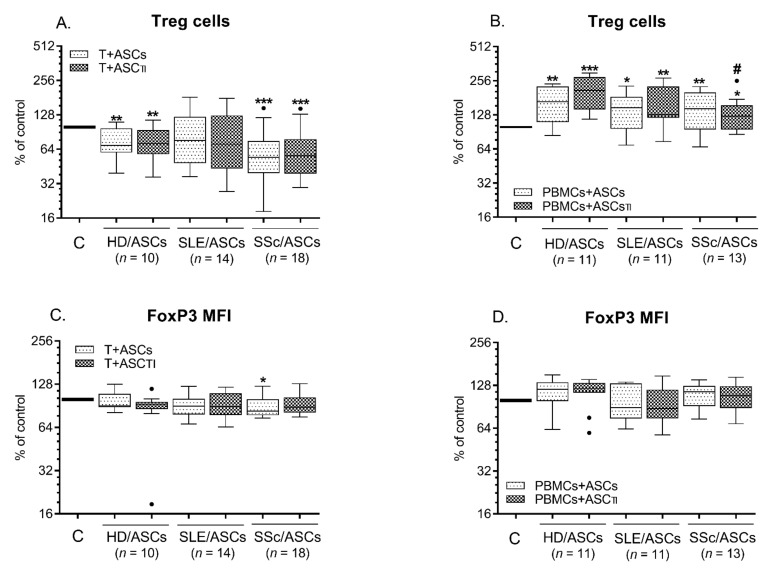
The generation of Tregs in the co-cultures of ASCs with activated target cells. CD4^+^ T cells and activated PBMCs. Cells were prepared and co-cultured as described in the Materials and methods. For the co-cultures of ASCs with activated T cells (**A**,**C**), CD4^+^ T lymphocytes isolated from peripheral blood of 13 healthy volunteers were randomly combined with 5 HD/ASCs, 10 SLE/ASCs, and 12 SSc/ASCs lines, while for the co-cultures of ASCs with activated PBMCs (**B**,**D**), PBMCs, HD/ASCs, SLE/ASCs, and SSc/ASCs were obtained from 6, 5, 8, and 9 donors, respectively. The number of Tregs (**A**,**B**) and the mean fluorescence density (MFI) of FoxP3 (**C**,**D**) were estimated as described in the Materials and methods. The results are expressed as the percentage of the Tregs regarding those found in separately cultured activated control target cells (**C**) and are shown as the Tukey’s boxes. * *p* = 0.05–0.01; ** *p* = 0.01–0.001; *** *p* = 0.001–0.0001 for intra-group comparison; # *p* = 0.05–0.01 for intra-group comparison; n—the number of experiments performed.

**Table 1 ijms-23-05317-t001:** Demographic and clinical characteristics of the patients.

	Systemic Lupus Erythematosus (SLE) (*n* = 19)	Systemic Sclerosis (SSc) (*n* = 19)
Demographics		
Age, years	43 (25)	49 (17)
Sex, female (F)/male (M), n	17F/2M	13F/6M
BMI	24.5 (11.5)	24.4 (3.4)
Disease duration, years	8 (10)	5 (8) ^a^/3 (5.5) ^b^
Clinical data		
Disease activity, score	6 (12) *	1 (1.9) **
Laboratory values		
CRP, mg/L	5 (9.5)	7 (5.5)
ESR, mm/h	17 (18)	17 (15)
Proteinuria, mg/24 h	65 (398) ^#^	0 (0.07) ^#^
C3, mg/dL	80.1 (41.9)	97.1 (24.7)
C4, mg/dL	15.5 (12.2)	16.6 (4.7)
ANA, titre (1:x)	1280 (3200) ^##^	5120 (5120) ^##^
anti-dsDNA antibody, %	84.6	n/a
anti-dsDNA antibody, IU/mL	54.6 (220)	n/a
Scl-70 antibody, %	n/a	64.7
Autoantibody specificities, no.,	2 (2.5)	2 (1)
Medications, %		
Immunosuppressive drugs	100	100

Except where indicated otherwise, values are the median (IQR). BMI, body mass index; duration of ^a,b^ Raynaud’s syndrome or skin/organ symptoms; * SLEDAI, SLE Disease Activity Index, ** EUSTAR, the European League Against Rheumatism Scleroderma Trials and Research revised index; CRP, C-reactive protein; ESR, erythrocyte sedimentation rate; C, complement components; ANA, antinuclear antibody; Scl-70, anti-topoisomerase I antibody; h, hour; n/a, not applicable. **^#^***p* = 0.05–0.01, and **^##^***p* = 0.01–0.001 for SLE vs. SSc comparisons.

## Data Availability

The data presented in this study are available in article’s figures and tables.

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
