# Peer review of "Impact of Adipose-Derived Mesenchymal Stem Cells (ASCs) of Rheumatic Disease Patients on T Helper Cell Differentiation"

_ijms, 2022, doi:10.3390/ijms23105317_

Round 1
Reviewer 1 Report
The manuscript by Kuca-Warnawin et al. “Impact of adipose-derived mesenchymal stem cells (ASCs) of rheumatic disease patients on T helper cell differentiation” is focused on the immunomodulatory properties of adipose-derived mesenchymal stromal cells isolated from patients with either systemic lupus erythematosus or systemic sclerosis, two autoimmune disease associated with chronic inflammation and high mortality rate. In particular, the authors focused on the ability of ASCs to restore the balance between Th1/Th2/Th17/Tregs Th-cell populations, which has been found dysregulated in both pathologies. Authors suggest that ASCs from patients perturb the transcription of lineage-specific Th1 and Tregs factors, without however a functional consequences, while a proper demonstration lacking, and thus conclude that ASCs from SLE and SSc patients have comparable immunomodulatory properties with ASCs from healthy donors.
General comments:
1) Results have been obtained with ASCs from several SLE and SSc patients, however authors utilize cells from patients undergoing immunosuppressive therapy. Although they recognize this limitation, they state that the effect of drug is vanished by a month-long expansion of the cells in vitro. I wonder if this long-term expansion impacts on senescence of cells. Have control HD cells undergone the same long-term expansion? It is unclear since in the methods section author state that cells have been used from p3 to p5. Does cells isolated from SLE and SSc patients growth at different rate than ASCs purchased for control? How are the expressions of staminal markers? After this long-term expansion do ASCs maintain the ability to differentiate?
2) How authors separated Th-cells from ASCs after co-culture to isolate mRNA? And how they normalize gene expression? Seems that they just control fold change of gene of interest without any normalization for housekeeping.
3) Does ASCs from patients have a different expression of anti-inflammatory cytokines compared to HD ASCs? Does INFγ/TNFα pre-treatment have different effects on ASCs from patients compared to HD?
Reviewer 2 Report
Comments to authors:
The authors are experts in their area, and in this article provide a detailed evaluation of the immune response of peripheral adipose tissue-derived cells (ASCs) from healthy donors, patients with lupus, and patients with sclerosis. These diseases are complex, and the authors do an admirable job parsing out observations about differences in the way diseased patient ASCs impact Th1 and Th2 immune activation responses. They uncovered interesting alterations in Th differentiation with sclerosis and lupus ASCs which was disturbed at the transcription level, but noted no functional consequence that resulted. The similarities between healthy and diseased sources for ASCs in Th1 to Th2 differentiation switching, the similarities in IFNγ/IL-4 ratio decrease, the up-regulated Th17 formation and IL-17 secretion, and the up-regulation of classical Tregs all support the potential future development of autologous ASC therapies even from patients with autoimmune diseases. The work is a good fit for the readership of the International Journal of Molecular Sciences. In addition to being technically sound the manuscript is well written, and in the eyes of this reviewer after addressing the minor revisions below the article is suitable for publication.
Comments:
-Lines 58-60. Clinical trials of allogeneic cells are mentioned. Please provide reference for current clinical trial numbers, or a recent relevant review articles.
-Line 61 “emphasise” —> “emphasize”
-Lines 64-65 “However, allogeneic MSCs can induce an adaptive immune response under appropriate conditions, followed by a significant adverse response.” This is a pretty strong statement, please provide evidence. In the eyes of this reviewer, MSC clinical trials are notable for their lack of adverse events.
Round 2
Reviewer 1 Report
The authors addressed at least in the letter the raised issues.